# Effective Preparation of [^18^F]Flumazenil Using Copper-Mediated Late-Stage Radiofluorination of a Stannyl Precursor

**DOI:** 10.3390/molecules27185931

**Published:** 2022-09-13

**Authors:** Mohammad B. Haskali, Peter D. Roselt, Terence J. O’Brien, Craig A. Hutton, Idrish Ali, Lucy Vivash, Bianca Jupp

**Affiliations:** 1Sir Peter MacCallum Department of Oncology, The University of Melbourne, Melbourne, VIC 3010, Australia; 2The Radiopharmaceutical Research Laboratory, The Peter MacCallum Cancer Centre, Melbourne, VIC 3000, Australia; 3Department of Neuroscience, Central Clinical School, Monash University, Melbourne, VIC 3004, Australia; 4School of Chemistry, The University of Melbourne, Melbourne, VIC 3010, Australia; 5Bio21 Molecular Science and Biotechnology Institute, The University of Melbourne, Melbourne, VIC 3010, Australia

**Keywords:** radiofluorination, PET imaging, GABA_A_, radiochemistry, Flumazenil 1, benzodiazepine, stannyl, stannane

## Abstract

(1) Background: [^18^F]Flumazenil **1** ([^18^F]FMZ) is an established positron emission tomography (PET) radiotracer for the imaging of the gamma-aminobutyric acid (GABA) receptor subtype, GABA_A_ in the brain. The production of [^18^F]FMZ **1** for its clinical use has proven to be challenging, requiring harsh radiochemical conditions, while affording low radiochemical yields. Fully characterized, new methods for the improved production of [^18^F]FMZ **1** are needed. (2) Methods: We investigate the use of late-stage copper-mediated radiofluorination of aryl stannanes to improve the production of [^18^F]FMZ **1** that is suitable for clinical use. Mass spectrometry was used to identify the chemical by-products that were produced under the reaction conditions. (3) Results: The radiosynthesis of [^18^F]FMZ **1** was fully automated using the iPhase FlexLab radiochemistry module, affording a 22.2 ± 2.7% (*n* = 5) decay-corrected yield after 80 min. [^18^F]FMZ **1** was obtained with a high radiochemical purity (>98%) and molar activity (247.9 ± 25.9 GBq/µmol). (4) Conclusions: The copper-mediated radiofluorination of the stannyl precursor is an effective strategy for the production of clinically suitable [^18^F]FMZ **1**.

## 1. Introduction

Flumazenil **1**, marketed as Romazicon, is primarily used clinically for the treatment of a benzodiazepine overdose and for the reversal of the sedative effects of anesthesia [1,2]. Flumazenil **1** is a potent competitive antagonist at the benzodiazepine site of the gamma-aminobutyric acid (GABA) receptor subtype, GABA_A_. The amino acid GABA is the primary inhibitory neurotransmitter in the central nervous system (CNS) and functions to inhibit neuronal activity through its action at the GABA_A_ receptors [3]. In addition to binding to GABA, these ligand-gated ion channels also bind benzodiazepines, a class of psychoactive drugs with a core structure encompassing a fused benzene ring and a diazepine ring [4,5] which facilitates the action of GABA at this receptor.

Alterations in GABAergic function including at the GABA_A_ receptor is associated with a variety of neurological disorders including substance use disorder, schizophrenia, autism spectrum disorder, major depressive disorder and epilepsy [6,7,8,9,10,11]. Therefore, a quantitative assessment of GABA_A_ receptors in the brain using positron emission tomography (PET) can provide valuable information concerning the GABAergic function of a broad range of neurological and neuropsychiatric conditions [12,13].

PET represents the most selective and sensitive (pico- to nano-molar range) non-invasive molecular imaging technique for the quantification of receptor density and drug interactions in vivo [14]. PET utilizes biologically active drugs at tracer doses which are radiolabeled with short-lived positron-emitting radionuclides. Fluorine-18 is one of the most useful positron-emitting radionuclides with ideal properties for PET imaging. Therefore, F-18-labelled flumazenil **1** has evolved to be one of the most useful radiopharmaceuticals for the PET imaging of GABA_A_ receptors [13,15]. [^18^F]Flumazenil **1** ([^18^F]FMZ) is now well established in the clinical management of drug-resistant temporal lobe epilepsy (TLE) with excellent sensitivity and anatomical resolution [13].

[^18^F]FMZ **1** is commonly prepared by the nucleophilic aromatic substitution (S_N_Ar) of the nitro group on nitromazenil using ‘naked’ fluoride-18 (Figure 1) [16]. Considering that nucleophilic aromatic substitution by fluoride-18 ion does not proceed well unless it is activated by a strong electron withdrawing group in the *ortho*- or *para*-position, the effective production of [^18^F]FMZ **1** by S_N_Ar has been subjected to considerable optimization to proceed with a satisfactory RCY [17]. The electron withdrawing groups enable the radiofluorination of arenes by reducing the electron density at the target carbon and by resonance stabilization of the arising Meisenheimer complex in an addition-elimination mechanism. Therefore, improved methods for the preparation of [^18^F]FMZ **1** in high yield and under mild chemical conditions, affording a satisfactory chemical purity, has been the subject of over a decade of radiopharmaceutical research.

Windhorst and co-workers initially reported the radiosynthesis of [^18^F]FMZ **1** by the substitution of the nitro group of nitromazenil or by the isotopic exchange (F-19 substituted by F-18) of flumazenil **1** (Figure 1) [18]. The radiofluorination was performed at an elevated temperature (≥130 °C), a reduced pressure (≤0.8 bar) and with the use of 2,4,6-trimethylpyridine as solvent. These harsh conditions afforded a moderate (44% yield by isotopic exchange) to low yield (≤12% yield by nitro substitution). Furthermore, the method using an isotopic exchange afforded [^18^F]FMZ **1** with a low molar activity (the ratio of the quantity of F-18-labeled molecules to the total quantity of FMZ (i.e., containing either F-18 or F-19)) [19]. The preparation of [^18^F]FMZ **1** by isotopic exchange was further improved by Ryzhikov and co-workers by employing standard radiofluorination conditions, but the final product retained a low molar activity [20]. Halldin and co-workers optimized the radiofluorination conditions for the substitution of the nitro group of nitromazenil to afford a moderate radiochemical yield (30%) and a high molar activity of [^18^F]FMZ **1,** at a high temperature (160 °C for 30 min) [16]. These conditions have since become the method of choice for the preparation of [^18^F]FMZ **1,** with nitromazenil precursor becoming commercially available through specialized suppliers. However, we note that Schirrmacher and co-workers reported only a 2–5% overall radiochemical yield (RCY) when adopting these conditions [21]. Similarly, van Dam and co-workers reported only a 7 ± 2% overall RCY under these conditions, thus highlighting the limitations of this method [22]. Our own experience with this method has also led to 3–4% isolated yields of [^18^F]FMZ **1** [13,23,24]. Some of these variations in RCY could arise from inconsistencies in the measurement of RCY and by not taking into consideration the recovery of radioactivity from the systems that are in use. 

A major breakthrough in the radiofluorination of inert aromatic systems was first described by Pike and coworkers, using diaryliodonium salts to facilitate substitution reactions [25]. This method has been applied by Seok Moon and co-workers to prepare [^18^F]FMZ **1** in a high radiochemical yield (67.2 ± 2.7% decay-corrected) [26]. More recent developments in F-18 radiochemistry have paved the way for the preparation of a wide range of radiopharmaceuticals that were previously inaccessible. One key development relates to the copper-mediated late-stage radiofluorination of aryl boronates and aryl stannanes. The application of this method to the preparation of [^18^F]FMZ **1** from a boronic ester precursor has been disclosed (Figure 1), affording a ≤17% decay-corrected radiochemical yield (*n* = 2) [27]. The usefulness of this method for the preparation of [^18^F]FMZ **1** for clinical use has recently been demonstrated [28]. Importantly, this recent development has optimized the formation of [^18^F]FMZ **1** to over a 30–48% radiochemical yield (non-decay-corrected) when TBA-HCO_3_ was utilized as a base. One challenging aspect relates to the susceptibility of boronic esters to hydrolysis and protodeboronation under the column chromatography conditions [27]. The copper-mediated radiofluorination of boronic esters requires air for the oxidative transformation of Cu(II) to Cu(III) in the catalytic cycle that facilitates aromatic radiofluorination [17,27,28]. These limitations may reduce the usefulness of this method. 

In contrast, the copper-mediated late-stage radiofluorination of aryl stannanes does not require air to proceed, and the aryltrialkylstannanes are generally stable under chromatographic conditions. The preparation of [^18^F]FMZ **1** through the copper-mediated late-stage fluorination of a stannyl precursor has not been reported. This method introduces some potential advantages, including the amenability to prepare large amounts of a stannyl precursor (gram scale) and its simple purification using column chromatography, with potentially high radiochemical yields, high molar activity and simple automation processes. Herein, we report the synthesis of a stannyl precursor to flumazenil and the conditions for its radiofluorination. Furthermore, we translate these conditions using the iPHASE FlexLab radiochemistry module to fully automate the radiosynthesis of [^18^F]FMZ **1**. We also investigate the chemical nature of the side products formed in the reaction using mass spectrometry, and we report on other quality control aspects of the final formulated product and its suitability for clinical use.

## 2. Results

### 2.1. Synthesis of Stannyl Precursor **3**

Stannyl precursor **3** was synthesized by the palladium-catalyzed stannylation of bromomazenil **2** (Figure 2). The stannylation reaction proceeded smoothly to produce precursor **3** in good chemical yield (65%) after 6 h in toluene. The stannyl precursor **3** was purified using flash chromatography, affording a pure solid material ready for the radiofluorination reactions. 

### 2.2. Radiosynthesis of [^18^F]flumazenil **1**

With the pure stannyl precursor **3** in hand, we moved toward the optimization of the copper-catalyzed radiofluorination conditions to obtain [^18^F]FMZ **1** in good radiochemical yield (Figure 3). The optimal radiofluorination conditions using copper (II) triflate, pyridine and DMA were maintained as recommended by Scott and co-workers [29]. We investigated the effects of temperature and time on the yield of [^18^F]FMZ **1** and its formation. 

The [^18^F]FMZ **1** % formation was heavily dependent on the reaction temperature. At 80 °C only 10.2% [^18^F]FMZ **1** was observed after 10 min of reaction time, and the total radioactivity that was recovered from the analytical HPLC column was less than 30%. The low recovery from the analytical HPLC column indicates that the radiochemical yield of [^18^F]FMZ **1** may be even lower than was observed (10.2%), as the majority of the free F-18 ions may be retained in the column. Free F-18 is known to have a notorious affinity to silica-based HPLC columns at a low pH [30]. As such, it is essential to consider the % column recovery of the total radioactivity that was injected when determining the radiochemical yield by the integration of a radiochemical HPLC trace. 

Increasing the reaction temperature to 100 °C increased the radiochemical yield of [^18^F]FMZ **1** to 51.9%. The % recovery of the radioactivity from the HPLC column also increased to 60% at 100 °C, indicating significant improvements in the radiochemical formation of [^18^F]FMZ **1**. Increasing the reaction temperature to 120 °C further increased radiochemical yield to 58.1% and the % recovery of the radioactivity to 63%. However, increasing the reaction temperature further to 140 °C did not increase the radiochemical yield, but it did increase the % recovery of the radioactivity to 80%. This may indicate that the isolated yield of [^18^F]FMZ **1** maybe higher at 140 °C. However, we chose to continue to perform our optimization reactions at 120 °C to avoid the harsher conditions at 140 °C that can damage the reactor apparatus over time. 

We then explored the effect of reaction time on the radiochemical yield of [^18^F]FMZ **1**. Reducing the reaction time to 5 min reduced the radiochemical yield of [^18^F]FMZ **1** at 120 °C to 51.1%, with similar HPLC column recovery (63.6%). Increasing reaction time to 20 min also reduced the radiochemical yield of [^18^F]FMZ **1** to 51.2%, with a slight increase in HPLC column recovery (71.0%). As such, the optimal conditions for the copper-catalyzed radiofluorination of the stannyl precursor **3** to form [^18^F]FMZ **1** was identified to be at 120 °C for 10 min (Table 1). 

The optimal reaction conditions were than translated using the iPHASE Flexlab radiochemistry module to automate the full production process, including the F-18 isolation and workup, radiofluorination, HPLC purification and final formulation of [^18^F]FMZ **1**. [^18^F]FMZ **1** was produced using the iPHASE Flexlab module using the configurations that are presented in Table 2. The total automated production time was 80 min to produce the fully formulated [^18^F]FMZ **1** in a 22.2 ± 2.7% (*n* = 5) isolated yield with was decay-corrected to end of synthesis.

### 2.3. Chemical Characterization and Quality Control of [^18^F]FMZ **1** Formulation

The [^18^F]FMZ **1** was produced to meet the quality control specifications for F-18-labelled radiopharmaceuticals which are in line with the international standards. Furthermore, special reference was made to the N-[^11^C-methyl]flumazenil injection monograph (BP) when assigning the release criteria in Table 3. [^18^F]FMZ **1** was produced in high radiochemical purity (>98%). The ethanol that was used during the formulation of [^18^F]FMZ **1** was always below 10%, and the other residual solvents that were used throughout the synthesis (MeCN and DMA) remained below their respective limit, as presented in Table 3.

### 2.4. MS Analysis of By-Products Generated during the Radiosynthesis of [^18^F]FMZ **1**

Stannyl precursor **3** was completely consumed after the reaction at 120 °C for 10 min. An MS analysis of the crude reaction mixture indicated the formation of at least three possible by-products as well as a trace amount of the carrier flumazenil **1**. The major by-product that was formed was assigned as hydroxy-mazenil **4** with the minor by-products being assigned as des-fluoro-flumazenil **5** and the dimeric mazenil **6** (Figure 4). The MS analysis of des-fluoro-flumazenil **5** presented the expected *m/z* of 286 but there was also another *m/z* peak at 387.2 which we were unable to identify. The formation of by-products **4**–**6** was further supported by a tandem MS/MS analysis, presenting fragmentation profiles that are consistent with that of an authentic reference standard of flumazenil 1 (MS/MS spectra and observed fragments presented in the Appendix A).

### 2.5. PET Imaging Using [^18^F]flumazenil **1**

PET imaging using [^18^F]FMZ **1** produced by our current method demonstrated uptake and binding that was consistent with that which was previously observed in rodent studies of [^18^F]FMZ **1** PET, with uptake that was primarily concentrated in the cortices and hippocampi, with minimal uptake in the pons. (Figure 5). The average hippocampal B_max_ was 19.45 ± 1.5 pmol/mL, while the 1/K_D_ was 0.25 ± 0.03 pmol/mL (Figure 6), which is in line with the values that were acquired from our previous study in naïve rats [23].

## 3. Discussion

The successful production protocols for a given radiopharmaceutical are determined by many factors, including the amenability of a high-scale precursor synthesis, the efficacy and simplicity of the radiofluorination process and the suitability of the final product for its clinical use as determined by its quality control characteristics. The commonly employed method for the preparation of [^18^F]FMZ **1** through the nucleophilic substitution of a nitro leaving group suffers from a low yield and poor reproducibility. In our laboratory, this method generated [^18^F]FMZ **1** in less than 4% radiochemical yield and with a complex HPLC purification. Alternative methods have been investigated for the preparation of [^18^F]FMZ **1,** including its radiochemical synthesis from diaryliodonium salts and boronic esters [26,27,28]. The latter method suffers from the instability of the boronic ester precursor under chromatographic conditions, and the need for an aerated reaction vessel [27,28].

The preparation and radiofluorination of the stannyl precursor **3** overcomes many of these complexities. The synthesis of stannyl precursor **3** proceeded in one step from the bromo-precursor **2,** and the synthesis proved to be scalable (≥2 g), affording high purity precursor after a rapid flash column chromatography. Furthermore, precursor **3** has proved stable for over a year when it is stored at 4 °C. The radiofluorination of stannyl precursor **3** proceeded efficiently without the need for aeration or phase transfer catalysts (including kryptofix or tetrabutylammonium bicarbonate). We note that Scott and co-workers have also demonstrated the sensitivity of the Cu-mediated radiofluorination of stannyl precursors to the type of base/phase transfer catalyst that is used, and therefore, potassium triflate that is doped with potassium carbonate was used as the optimal combination [29]. Nonetheless, it is useful that no phase transfer catalyst is required under these conditions, thereby eliminating the need to test the end product for kryptofix or tetrabutylammonium before its clinical administration. Finally, [^18^F]FMZ **1** was obtained in high radiochemical purity after HPLC purification. Nevertheless, small closely eluting UV-active byproducts were formed, and the HPLC conditions needed to be optimized carefully to allow for the adequate separation of the product from the impurities. 

The analysis of the crude reaction mixture identified the formation of several chemical by-products which may be addressed in the future to further enhance the yield of [^18^F]FMZ **1,** as well as to improve the purification process. The stannyl precursor **3** was completely consumed in the reaction mixture and the hydroxylated by-product **4** was found to be the major by-product along with by the product of protodestannylation (affording des-fluoro-flumazenil **5**). Hydroxylated by-product **4** may arise through the oxidation of the stannane under the reaction conditions [31]. A small amount of biaryl by-product **6** was also formed through homocoupling [32,33]. However, considering that the yield of [^18^F]FMZ **1** was useful for multi-patient dose preparation, and the HPLC purification was successfully optimized, no further optimizations were performed. This new method of [^18^F]FMZ preparation will facilitate the more ready utilization of this highly selective and sensitive radiotracer for the GABA_A_ receptor in clinical practice, to assist in the localization of the epileptogenic zone in patients with drug resistant TLE [13], as well as other potential clinical applications where the dysfunction GABA_A_ receptors are believed to play a role such as traumatic brain injury, schizophrenia, addiction and anxiety.

## 4. Materials and Methods

All chemicals obtained commercially were of analytical grade and used without further purification. No-carrier-added fluoride-18 was obtained from a PETtrace 16.5MeV cyclotron (Cyclotek) incorporating a high-pressure niobium target via the ^18^O(p,n)^18^F nuclear reaction (98% ^18^O isotopic enrichment). Radiochemical synthesis was performed using an iPHASE Flexlab radiochemistry module purchased from iPHASE Technologies Pty. Ltd. Australia. F-18 Separation cartridges (QMA strong anion exchange cartridge, Waters Australia) were preconditioned with 0.5 mL of 0.05M solution of KOTf, followed by 5 mL water. Reversed phase solid phase extraction (SPE) cartridges (33 μm polymeric reversed phase (30 mg/mL), Phenomenex, Lane Cove West, NSW, Australia) were preconditioned with ethanol and rinsed with water before use. Radioactivity measurements were carried out using a CRC-15PET dose calibrator (Capintec, Florham Park, NJ, USA) that was calibrated daily using Cs-137 and Co-57 sources (Isotope Products Laboratories, Valencia, CA, USA). Radiation was detected using a solid-state photodiode scintillator crystal detector (Knauer, Berlin, Germany). Preparative high performance liquid radiochemical chromatography (HPLRC) was performed using a Knauer 1050 pump, 2500 UV detector, and 5050 manager. Radiation was detected using a Knauer solid state photodiode scintillation crystal detector in a TO-5 case. Analytical HPLRC was performed using a Shimadzu HPLC system consisting of a CBM-20A system controller, SIL-20A auto-injector, LC-20AD solvent delivery unit, CTO-20A control valve, DGU-20A degasser and a SPD-M20A detector coupled to a LCMS-8030 triple-quadrupole mass spectrometer. This was coupled to a radiation detector consisting of an Ortec model 276 photomultiplier base with a 925-SCINTACE-mate preamplifier, amplifier, bias supply, and SCA and a Bicron 1M11/2 photomultiplier tube. Gas Chromatography (GC) analysis was performed using a Shimadzu GC-17A instrument coupled with an AOC-20i auto injector. ^1^H NMR spectra of small molecules were obtained using a 400 MHz Agilent DD2 NMR Spectrometer. 

**Synthesis of ethyl 5,6-dihydro-5-methyl-6-oxo-8-tributylstannyl-4*H*-imidazo-[1,5-a]**[1,4]**benzodiazepine-3-carboxylate (3)**: To a solution of ethyl 8-bromo-5,6-dihydro-5-methyl-6-oxo(4H)-imidazo[1,5-a][1,4]benzodiazepine-3-carboxylate 2 (2.2 g, 6.04 mmol) in toluene (20 mL) was added tetrakis-(triphenylphosphine)palladium(0) (0.25 g, 0.22 mmol) and bis(tributyltin) (10.5 g, 18.2 mmol). The reaction was purged with nitrogen in a pressure vessel and then heated to 120 °C for 6 h. The reaction mixture was then cooled and diluted with ethyl acetate (300 mL) and washed with water, dried over anhydrous sodium sulfate, and evaporated to dryness in vacuo. Crude reaction mixture was purified by flash column chromatography starting with hexane (100%) to remove any unreacted bis(tributyltin)(R_f_ = 0.95), followed by the elution of stannyl precursor 3 using ethyl acetate (100%; R_f_ = 0.33). The title compound was isolated as an oil (2.25 g, 65%) that slowly crystalized after storage at 4 °C into a sticky white solid. ^1^H-NMR (400 MHz, CDCl_3_) δ 0.87 (t, J = 7.2 Hz, 9H), 1.1.03–1.19 (m, 6H), 1.28–1.36 (m, 6H), 1.43 (t, *J* = 7.2 Hz, 3H), 1.47–1.63 (m, 6H), 3.23 (s, 3H), 4.30–4.50 (m, 3H), 5.19 (m, 1H), 7.34 (d, *J* = 8.0 Hz, 1H), 7.69 (dd, *J* = 8.0, 1.2 Hz, 1H), 7.87 (s, 1H), 8.11 (d, *J* = 0.8 Hz, 1H). ESI-MS: *m/z* 576.2 [M^+^ + H]. These data match the literature data [26].

**Radiochemistry**: Fluoride-18 (33.3–74.0 GBq) was trapped on QMA cartridge and azeotropically dried according to our previously reported procedures using the iPHASE FlexLab radiochemistry module [34]. Potassium triflate:potassium carbonate (10 mg:0.05 mg; 550 µL) was used to elute F-18 from the QMA cartridge and prepare dried K[^18^F]F. To the dried K[^18^F]F, stannyl precursor 3 (5 mg, 8.7 μmol) in DMA (1 mL) containing copper, triflate:pyridine (14 mg:13 µL; 28.7 µmol:146 µmol) was added. After 10 min at 120 °C, the residue was diluted with 0.1% TFA H_2_O/MeCN (2.5:0.5, 3 mL). The mixture was then purified by preparative HPLC on a Kinetex 5 µm XB-C18 250 × 150 mm column, 0.1% TFA in 15–80% MeCN:H_2_O over 40 min. Isolated product was diluted in water (30 mL) and then trapped on a C18 SEP-PAK cartridge. The trapped product was rinsed with saline (5 mL), eluted with ethanol (1 mL) and diluted with saline (10 mL) before sterile filtration (Vented Cathivex GV, 0.22 μm, 25 mm) into a sterile evacuated product vial (FILL-EASE^TM^ Sterile Vials; SVV-15A) was performed to prepare the title compound to be ready for injection (2–8 GBq, 22.2 ± 2.7% isolated yield (*n* = 5)). The total reaction time was 80 min.

During optimization reactions, a small fraction (10 µL) of the crude reaction mixture was diluted into water:MeCN (75:25, 90 µL) and directly injected onto an analytical HPLC system. The % recovery of radioactivity from the analytical HPLC column was quantified by accurately measuring total injected amount of radioactivity using a dose calibrator. Every radioactive peak was then collected from the waste line after passing the radioactivity detector. % recovery was then calculated as the following: (collected total radioactivity/injected total radioactivity) × 100.


**Sample preparation and extraction protocol for inductively coupled plasma-mass spectrometry (ICP-MS)**


For the quantitative analysis of Copper (Cu), 100 µL of each sample were digested with 50 µL concentrated 65% nitric acid (HNO_3_,70% Analytical grade from Ajax Finechem) followed by heating at 95 °C for 10 min. The samples were then diluted with Milli-Q water (18.2 MΩ; Milli-Q H_2_O; Merk Millipore, Australia) (1:10 to a final volume of 1 mL). Samples were briefly vortexed and centrifuged at 15,000 rpm for 25 min and supernatant was immediately transferred to new 1.7 mL microcentrifuge tubes. Sample blanks were prepared in the same manner.


**Inductively coupled plasma-mass spectrometry (ICP-MS)**


Tuning solution containing 1 μg/L of cerium (Ce), cobalt (Co), lithium (Li), thallium (Tl) and Y in 2% (*v*/*v*) HNO_3_ (Agilent Technologies, Australia) was used to tune and optimize the Agilent 8900 triple quadrupole ICP-MS (Agilent Technologies, Australia) in a Helium gas analysis mode. A 9-point calibration (0, 1, 5, 10,25, 50, 100, 250 and 500 parts per billion (ppb) in 1% HNO_3_) standard curve for Cu was prepared using commercially available multi-element standards (Multi-Element Calibration Standard 2A, Agilent Technologies, USA). The R^2^ value for copper calibration curve was 0.999617. ICP-MS analysis method for Cu detection yielded a limit of detection (LOD) of less than 0.0993 μg/L and a limit of quantitation (LOQ) around 0.33 μg/L. Yittrium (^89^Y) (Agilent Technologies, USA) was used as an internal standard at a concentration of 0.1 μg/mL and used as reference element solution to normalize all measurements. All the samples, calibration standards and reference solution were introduced at the flow rate of 0.4 mL/min using a T-piece and a peristaltic pump. The data was collected in spectrum mode with the average of three technical replicates. The ICP-MS operating parameters were established according to the manufacturer’s guidelines and other parameters were optimized for copper in a batch-specific mode prior to each experiment and these are as follows: ICP-MS operating parameters; Scan type: Single Quad, RF Power: 1550W, RF Matching: 1.8V, Nebulizer Gas: 1.05L/min, Extract 1: −12V, Extract 2: 250V, Omega Bias: −120V, Omega Lens: 7.2V, Deflect: −5V, Gas: He Gas, Oct P bias: −18V.

**Small animal PET imaging**: Six adult, male Wistar rats (Monash Animal Resources Centre, 0.500 ± 0.071 kg) were anesthetized with isoflurane (induction: 5% in 1 L/min O_2_, maintenance 1.5–2% in 1 L/min O_2_) and [^18^F]flumazenil (dose: 23.5 ± 10 MBq, mass: 3.6 ± 1.2 nmol) was injected as a bolus over 10s via the dorsal penile vein, as previously described [23]. Immediately following tracer injection, dynamic PET scans were acquired for 45 min on a nanoScan-PET/CT (Mediso, Hungary). PET images were reconstructed across the following time frames (2 × 30 s, 2 × 60 s, 14 × 180 s) using the Tera-tomo 3D algorithm provided by the supplier with correction for scatter, attenuation and dead-time.

PET scans were manually co-registered to a corresponding T2-weighted MRI acquired from each rat acquired using a 9.4T Bruker Avance IIIHD MRI (Bruker, Germany) with actively decoupled 4-channel receive-only surface and volume transmit coils (voxel size 0.1 × 0.1 × 0.7 mm^3^; matrix 256 × 256 × 24) using ITK-SNAP [35], and resliced using linear interpolation. Volumes of interest (VOI) incorporating both left and right hippocampus and pons were manually delineated on the MR to extract time activity curves from the PET scans in PMOD (PMOD Technologies, Switzerland). B_max_ (receptor number) and 1/K_D_ (receptor affinity) were estimated from each VOI using a nonlinear fit of the bound ligand (B) versus the free ligand (F) (activity in pons): B_max_ = (F × B)/(F + K_D_) in Prism. A representative image is provided in Figure 5.

The study was conducted in accordance with the Australian NH&MRC Code of conduct for use of animals in research and the study protocol was approved by the Alfred Research Alliance Animal Ethics Committee (E/2004/2020/M).


**IUPAC nomenclature:**


*Flumazenil* ***1***: Ethyl 8-fluoro-5-methyl-6-oxo-5,6-dihydro-4*H*-benzo[*f*]imidazo[1,5-*a*][1,4]diazepine-3-carboxylate

*Stannyl precursor **3***: Ethyl 5-methyl-6-oxo-8-(tributylstannyl)-5,6-dihydro-4*H*-benzo[*f*]imidazo[1,5-a][1,4]diazepine-3-carboxylate

*Hydroxylated by-product **4***: Ethyl 8-hydroxy-5-methyl-6-oxo-5,6-dihydro-4*H*-benzo[*f*]imidazo[1,5-a][1,4]diazepine-3-carboxylate

*Des-fluoro-flumazenil **5***: Ethyl 5-methyl-6-oxo-5,6-dihydro-4*H*-benzo[*f*]imidazo[1,5-a][1,4]diazepine-3-carboxylate

*Homodimerization product **6***: 8,8′-Bis[Ethyl 5-methyl-6-oxo-5,6-dihydro-4*H*-benzo[*f*]imidazo[1,5-*a*][1,4]diazepine-3-carboxylate].

## Figures and Tables

**Figure 1 molecules-27-05931-f001:**
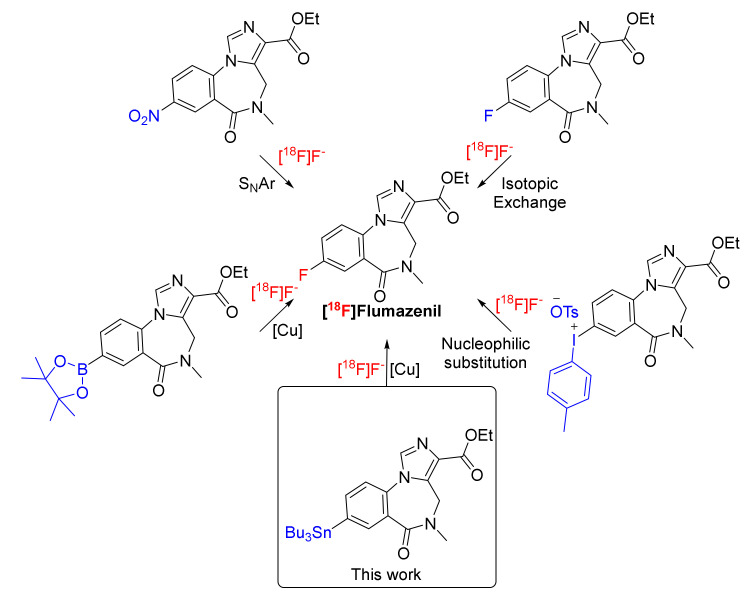
Nucleophilic radiofluorination methods for the preparation of [^18^F]flumazenil **1**.

**Figure 2 molecules-27-05931-f002:**
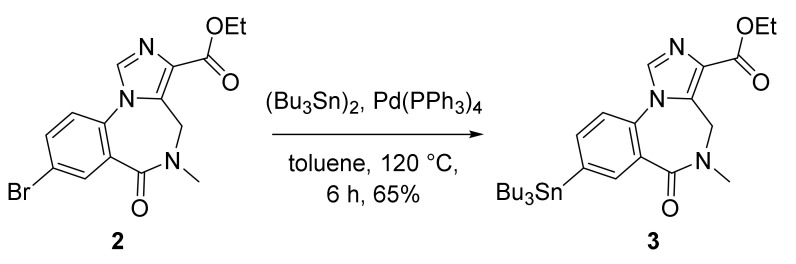
Synthesis of stannyl precursor **3**.

**Figure 3 molecules-27-05931-f003:**
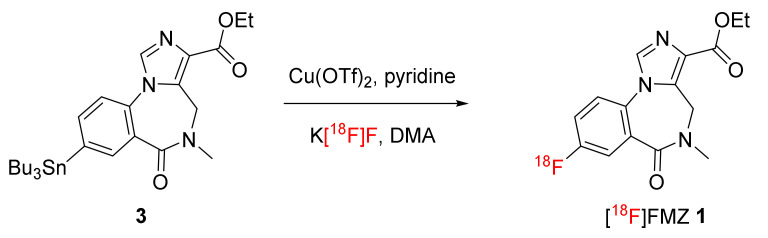
Cu-catalyzed radiosynthesis of [^18^F]FMZ **1**.

**Figure 4 molecules-27-05931-f004:**
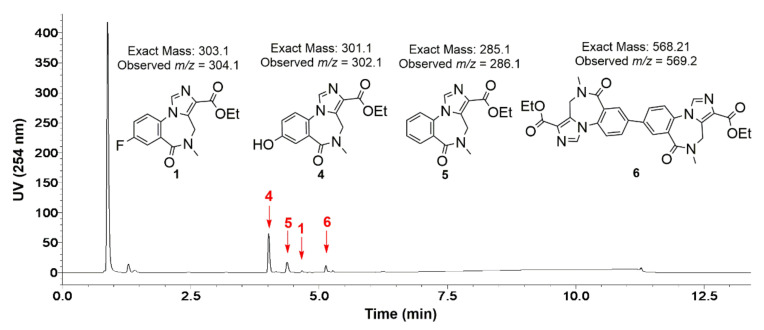
LC-MS/MS analysis of major by-products that were formed in the crude reaction mixture of [^18^F]FMZ **1** at 120 °C after 10 min reaction time.

**Figure 5 molecules-27-05931-f005:**
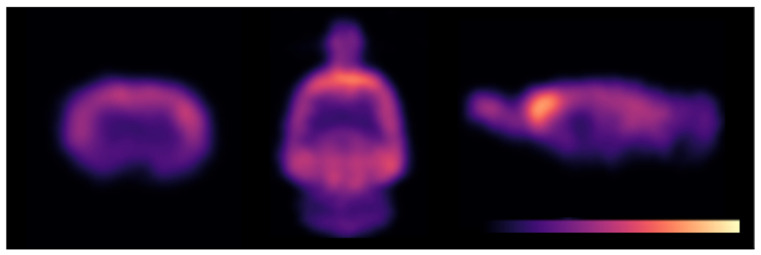
Representative image of [^18^F]flumazenil **1** uptake.

**Figure 6 molecules-27-05931-f006:**
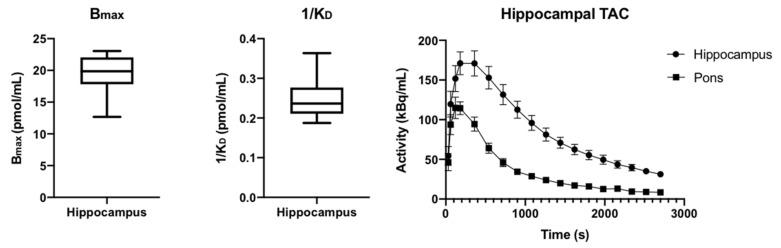
Box and whisker plots of hippocampal B_max_ and 1/K_D_ values and average time activity curves (TAC) from hippocampal and pons volumes of interest.

**Table 1 molecules-27-05931-t001:** Optimization conditions for the production of [^18^F]FMZ **1**.

Entry ^1^	Temperature (°C)	Time (min)	[^18^F]FMZ RCY% ^2^	HPLC Column Recovery (%) ^3^
1	80	10	10	29.7
2	100	10	52	60.0
3	120	10	58	63.0
4	140	10	57	80.0
5	120	5	51	63.6
6	120	20	51	71.0

^1^ Radiolabeling conditions: 10 µmol stannyl precursor **3**, 40 µmol Cu(OTf)_2_, 150 µmol pyridine in 1 mL DMA. ^2^ Radiochemical yield determined by integration of the HPLC radiochemical trace obtained from the analytical HPLC analysis of the crude reaction mixture. Reactions were performed once under each of these conditions. HPLC analysis was performed on Kinetex 5 µm XB-C18 4.6 × 150 mm column, 0.1% TFA in 15–90% MeCN:H_2_O over 7 min. ^3^ HPLC column recovery (%) obtained by accurately measuring total injected amount of radioactivity and the total collected radioactive eluted from the HPLC column. % Recovery was calculated as following: (collected total radioactivity/injected total radioactivity)*100. HPLC analysis was performed on Kinetex 5 µm XB-C18 4.6 × 150 mm column, 0.1% TFA in 15–90% MeCN:H_2_O over 7 min.

**Table 2 molecules-27-05931-t002:** Preparation details for the automated production of [^18^F]FMZ **1** using the iPHASE Flexlab radiochemistry module.

Entry	Position	Reagents or Materials	Quantities
1	V 13–V 15	Sep-Pak Light QMA	1
2	V 1	KOTf_/_ K_2_CO_3_ in H_2_O	10 mg KOTf and 50 μg of K_2_CO_3_ in 550 µL H_2_O
3	V 4	Stannyl Precursor **3** and Cu(OTf)_2_/Pyridine mixture	5 mg of precursor **3** in 1.0 mL DMA containing 13 mg of Cu(OTf)_2_ and 14 µL of pyridine
4	V 6	0.1% TFA in MeCN:H_2_O	3.0 mL (2.5:0.5)
5	V 8	Saline	9 mL
6	V 9	Ethanol	1 mL
7	V 11	Saline	5 mL
8	HPLC Flask 1	Milli-Q Water	50 mL
9	V 22–V 46	C18 SPE Cartridge	1

**Table 3 molecules-27-05931-t003:** Quality control specifications of [^18^F]FMZ **1**.

Parameter	Specification	Observed Results (*n* = 3)
Appearance	Clear and colorless	Pass
pH	4–8	5–6
Residual Solvents (MeCN) (%*V*/*V*)	˂0.04	0.0066% ± 0.0018% (0.006–0.008%)^3^
Residual Solvents (DMA) (%*V*/*V*)	˂0.11	Not detected
Ethanol Determination (%*V*/*V*)	˂10%	7.16% ± 1.38% (6.1–8.1%)^3^
Radionuclidic identity (half-life)	105–115 min	108–113
Radiochemical identity (HPLC)	Reference standard ± 1.0 min	Reference Std: 8.5 min Product: 8.6 min
Radiochemical Purity (HPLC)	≥95%	>98%
Radiochemical Purity (TLC)	≥98%	>98%
Molar Activity	≥37 GBq/µmol	247.9 ± 25.9 GBq/µmol (222–274 GBq/µmol)
Copper *	≤34 ppm	0.0157 ± 0.005 ppm
Sterility	Sterile	No growth observed
Endotoxin	≤175 IU/V	Pass
Filter Integrity (bubble point test)	≥50 psi	Pass

* Copper was quantified by inductively coupled plasma-mass spectrometry (ICP-MS) of the non-radioactive product after complete decay of F-18.

## Data Availability

Appendix A is available.

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
