# Peer review of "Effective Preparation of [18F]Flumazenil Using Copper-Mediated Late-Stage Radiofluorination of a Stannyl Precursor"

_molecules, 2022, doi:10.3390/molecules27185931_

Round 1

Reviewer 1 Report

Jupp et al. present herein the effective preparation of 18F-FMZ using a new late stage raàdiofluorination of a stannyl precursor. The experiments range from the synthesis, radiolabelling manually and automatically and in vivo evaluation. The results are well presented and sounds correct.

The reviewer has some doubt about the attribution of the byproduct formed during the radiosynthesis. Providing LR-MS is not suitable enough to elucidate the chemical structure of these byproducts. The authors must isolate these compounds and characterized them by ESI-HRMS, NMR, HPLC and should compared these to synthesized ones. For instance it is hard to trust the formation of as a major product when looking at the MS analysis. What is the compound affiliated to the predominant peak at m/z = 387.20?

The manuscript deserve to be publish after improvement. However the reviewer is a bit ashamed with the chemical errors made throughout the text, chemical formula, units (very important), etc... these errors shouldn't be present in a submitted manuscript from highly qualified researchers and corrected before acceptance. 

For instance, Fig 2, Pd(PPh3) is incorrect. 

Table 2. uL must be changed

Fig 4. nm instead of nM

L211 pmol/ml should be changed by pmol/mL

265 ml should be changed by mL

316-317 ul and HNO3 should be corrected.

This is not an exhaustive list of all the mistakes found in the manuscript that need proper proof-reading and correction. 

Reviewer 2 Report

The manuscript of Haskali et al. describes the preparation of [18F]flumazenil via Cu-mediated radiofluorination of the corresponding trimethylstannyl precursor and implementation of the developed procedure to the iPhase FlexLab radiosynthesis module. This paper is of interest for the radiochemical community and could be of some interest for the general readership of the Journal. Provided my concerns listed below will be adequately addressed, I will be happy to recommend the revised paper for publication.

Major comments:

1.       The description of previous [18F]FMZ production protocols is in part misleading. Whereas the critical review of the procedures for the preparation of [18F]FMZ via isotopic exchange and from the respective nitro-precursor is reasonable, the evaluation of [18F]FMZ radiosyntheses from the corresponding iodonium salt and pinacol boronate precursors is simply incorrect.

To date, the production of [18F]FMZ from the corresponding (4-methylphenyl)iodonium salt described by Moon et al. is one of the most efficient route to this tracer (RCY = 53±9%, n = 94, including production failure; ref. 24 and 10.1007/s11307-014-0738-z). The authors write (p.2, lines 92, 93): “Unfortunately, this method has not been adopted by other laboratories, making its reproducibility and tolerance unknown.” One should not question the reproducibility of the described procedure only because it “has not been adopted by other laboratories”. The authors should either demonstrate the irreproducibility of the radiosynthesis or remove this point from the manuscript.

The authors also state (p. 2, lines 93, 94): “Furthermore, the preparation of diaryl iodonium salts can be challenging to the non-familiar operator, which we believe has restricted the wide applicability of this method.” Whereas any organic synthesis could be challenging for “the non-familiar operator”, the desired iodonium salt precursor is easily accessible from the corresponding trialkylstannyl precursor in a simple single step (ref. 24). The authors should remove this claim from the manuscript.

Additionally, the authors assert (p. 8, lines 228, 229): ”The stability of the diaryliodonium salt precursor may be low and its chemical synthesis maybe complex thereby limiting the utility of this protocol.” The stability evaluation of the flumazenil (4-methylphenyl)iodonium salt precursor is provided in ref. 24. Accordingly, the precursor was sufficiently stable even at 150 oC in DMF in the presence of K2CO3/K2.2.2 and TEMPO (approx. 80% intact after 15 min). The authors should remove this assertion from the manuscript.

According to the authors (p. 3, lines 99, 100), RCYs of [18F]FMZ via Cu-mediated radiofluorination of the corresponding Bpin precursor did not exceed 17%. This statement is misleading because even though application of K2C2O4/K2CO3 for elution of [18F]F from an anion exchange resin indeed led to RCYs of ≤17% (ref. 25), it has since been shown that the application of TBA-HCO3 for this purpose increases the RCY to 48% (ref. 26). Please, remove or correct this statement.

The authors claim (p. 3, lines 102, 103): “One challenging aspect relates to the susceptibility of boronic esters to hydrolysis and protodeboronation under column chromatography conditions [25].” Indeed, according to SI of ref. 25, Bpin precursor of [18F]FMZ was not completely stable on silica. Nevertheless, this substance was prepared in 52% yield, which is well comparable with that for the trimethylstannyl precursor (65%). Furthermore, decomposition of the Bpin precursor on silica could be completely or at least significantly suppressed if silica with 0.1% Ca (accessible from Aldrich) or silica deactivated with Et3N would be used. Accordingly, this point is not relevant for the article and should be corrected or removed (also in the Introduction).

Next, the authors guess (p. 3, lines 232–235): “the need for high precursor amounts (12 mg) to facilitate effective radiofluorination, the need for aerated reaction mixture and the sensitivity of the radiofluorination conditions to the type of base used (potassium oxylate and tetrabutylammonium bicarbonate being optimal) [25, 26]” represent further significant disadvantages of the production of [18F]FMZ from the Bpin precursor. However, using 12 mg precursor, Gendron et al. (ref. 26) prepared [18F]FMZ in 48% RCY, which more than two times higher than obtained by the authors with 5 mg trimethylstannyl [18F]FMZ precursor (22%). Hence, lowering of Bpin precursor amount to 5 mg should still afford [18F]FMZ in approx. 20% RCY. Aeration of the reaction mixture could be very simply achieved if synthetic air is used as an operating gas. Lastly, Cu-mediated radiofluorination of stannanes is also sensitive to the type of base used (the authors apply KOTf doped with K2CO3). This fragment should be completely revised or removed (also similar fragment in the Introduction).

2.       Description and methodological flows.

From the manuscript it is even unclear, in which scale [18F]FMZ was produced?

It also remains unclear whether the individual optimization experiments were carried out once or multiple times. If they were performed only once, they do not provide any information on the reproducibility of the results. In any case, all RCYs should be rounded to whole numbers.

Despite the statement “free F-18 is known to have notorious affinity to silica-based HPLC columns at low pH” (page 4, lines 148, 149; here ref: 10.1016/j.jpba.2015.04.009 should be added), the authors applied a TFA-containing mobile phase, which heavily complicates the determination of RCYs by HPLC. The reason for this remains unclear, because acid-free eluents were used for [18F]FMZ analytics and isolation (refs 24 and 26).

Minor comments:

1.       Mazenil is not a commonly accepted name for des-fluoro-flumazenil. Please, correct “stannyl-mazenil” to, e.g., “stannyl precursor 3” throughout the whole manuscript.

2.       Abstract: “[18F]FMZ 1 was obtained in high radiochemical purity and molar activity”. Please provide quantitative data for radiochemical purity and molar activity in the abstract and in the manuscript and the appropriate quality control HPLC traces in the Supporting Information. Please, provide the description of the molar activity determination, including calibration curves.

3.       P.2, line 45: “Non-invasive PET”. Please, provide the full name for the abbreviation. Do you know “invasive PET”? If not, please, correct.

4.       P. 2, line 62: “under tolerable chemical conditions”. What the authors mean? Please, specify.

5.       P. 3, lines 109, 110: “the stannyl precursors are generally stable under chromatographic conditions”. It is generally not correct. Please, revise to, e.g., “aryltrialkylstannanes are usually stable on silica”.

6.       P. 4, line 136: “conidtions“. Please, correct.

7.       P. 4, line 137: “radioflourination”. Please, correct.

8.       P. 4, lines 160, 161: “harsh conditions at 140 °C that may volatilize pyridine and DMA”. DMA boils at 165 oC. Please, revise.

9.       P. 5, line 179: “where” should be changed to “were”.

10.   P. 8, line 228: “radiochemical synthesis of diaryliodonium salts and boronic esters” should be changed to “radiochemical synthesis from diaryliodonium salts and boronic esters”.

11.   P. 8, lines 241, 242: “or phase transfer catalysts (including kryptofix or tetrabutylammonium bicarbonate).” TBA-HCO3 is used in ref. 26 as base and not as phase transfer catalyst. Please, revise.

12.   P. 9, lines 299–301: “Fluoride-18 was prepared, trapped on QMA cartridge and azeotropically dried according to our previously reported procedures using the iPHASE FlexLab radiochemistry module.” Did you prepare 18F in the radiosynthesis module? Or in cyclotron? Please, correct. Please, provide the appropriate reference regarding 18F preprocessing.

13.   Figure 1: “[18F]” should be changed to “[18F]F“.

14.   Figure 3: “K18F” should be changed to “K[18F]F”.

Reviewer 3 Report

Authors developed an effective production of [18F]Flumazenil by Cu-mediated 18F-fluorination using an arylstannane precursor, which met the radiopharmaceutical qualities applicable to clinical PET study.

In my opinion, authors’ manuscript does not include any scientifically innovative work. Most of their radiolabeling experiments were based on the previous chemical reports of fluorination reactions.

However, I understand that authors’ work is very important from the viewpoints of PET radiochemistry for human use, and I know that there are not so much suitable chemistry journals to report authors’ sincere efforts for radiopharmaceutical production by aromatic 18F-fluorination.

From the comprehensive standpoints, I would like to recommend this paper for publication in Molecules.

There are some significant issues that should be addressed to increase the article quality from the academic perspective.

Regarding side-products of hydroxy-mazenil 4, destannylated mazenil 5, and dimeric-mazenil 6, authors should mention the chemical processes or reasons to generate such by-products from the viewpoints of organometallic chemistry.

I hope that authors reconfirm the IUPAC names of their new compounds, concerning the use of italic, hyphen, etc.

Round 2

Reviewer 1 Report

The authors are now presenting a revised version of the manuscript which takes into account some of the reviewers comments and the proof reading has been done carefully.

However the authors must provide clear evidence for the formation of the byproducts. If they performed some LC-MS/MS analysis, which the reviewer do not doubt and clearly acknowledges, the results must be presented and more detailed in the manuscript and supporting informations. Also the reviewer is still wondering what the predominant peak for the formation of (m/z = 387.20) is related to and should be explained.  

Author Response

MS/MS spectras and the respective structures of fragmented ions are now presented in the supporting data file.  

As for the peak appearing at m/z 387.20, we are unable to discern its identity. This has now been clarified in the manuscript in the following paragraph: 

"MS analysis of the crude reaction mixture indicated the formation of at least three possible by-products as well as a trace amount of the carrier flumazenil 1. The major by-product formed was assigned as hydroxy-mazenil 4 with the minor by-products assigned as des-fluoro-flumazenil 5 and the dimeric mazenil 6 (Fig. 4). MS analysis of des-fluoro-flumazenil 5 presented the expected m/z of 286 but there was also another m/z peak at 387.2 which we were unable to identify. The formation of by-products 46 was further supported by tandem MS/MS analysis presenting fragmentation profiles that are consistent with that of an authentic reference standard of flumazenil 1 (MS/MS spectra and observed fragments presented in the supplementary data file)."

Reviewer 2 Report

The authors have substantially improved the paper. However, a few points remain be revised before publication.

Major point:

Automated radiosynthesis is not described in sufficient detail to enable its successful reproduction.

Please, add at least the process flow diagram for the automated production of [18F]flumazenil.

P. 9, 10, lines 307-309: “Fluoride-18 (33.3–74.0 GBq) was trapped on QMA cartridge and azeotropically dried according to our previously reported procedures using the iPHASE FlexLab radiochemistry module.” Please, add the appropriate citation or describe processing of fluoride-18 in detail.

Minor points:

P. 2,3, lines 97-99: “Importantly, this recent development has optimized the formation of [18F]FMZ 1 to over 30 % radiochemical yield (non-decay corrected) when TBA-HCO3 was utilized a base.” Please, correct “over 30 %” to "48 %".

P. 9, line 302: “0..87”. Please, revise.

Author Response

1- Table 2 in the manuscript presents detailed conditions for automation but we has also provided the automated recipe as an attachment so that other's can adapt it easily

2- Citation (ref 34) added to P. 9, 10, lines 307-309

3- P. 2,3, lines 97-99: corrected as recommended by the reviewer. 

4- P. 9, line 302: “0..87” has been revised and one full stop has been deleted. 

Reviewer 3 Report

I think that the revised manuscript will meet the high quality of Molecules.

Author Response

N/A